# Using the Theory of Perceived Value to Determine the Willingness to Consume Foods from a Healthy Brand: The Role of Health Consciousness

**DOI:** 10.3390/nu16131995

**Published:** 2024-06-23

**Authors:** Roger Albornoz, Elizabeth Emperatriz García-Salirrosas, Dany Yudet Millones-Liza, Miluska Villar-Guevara, Gladys Toyohama-Pocco

**Affiliations:** 1Unidad de Ciencias Empresariales, Escuela de Posgrado, Universidad Peruana Unión, Lima 15102, Peru; roger.albornoz@upeu.edu.pe (R.A.); dannie@upeu.edu.pe (D.Y.M.-L.); gladystoyohama@upeu.edu.pe (G.T.-P.); 2Faculty of Management Science, Universidad Autónoma del Perú, Lima 15842, Peru; 3Escuela Profesional de Administración, Facultad de Ciencias Empresariales, Universidad Peruana Unión, Lima 15102, Peru; 4Escuela Profesional de Administración, Facultad de Ciencias Empresariales, Universidad Peruana Unión, Juliaca 21101, Peru; miluskavillar@upeu.edu.pe

**Keywords:** healthy foods, diet, healthy brand, health consciousness, perceived value

## Abstract

Eating low amounts of healthy foods leads to high rates of diet-related diseases. How can we control and reduce the increase in these diseases? One of the recommendations is to improve nutritional competence, which means greater health consciousness. The objective of the present study is to determine the influence of health consciousness on the dimensions of perceived value and their impact on the willingness to consume foods from a healthy brand. Through a non-experimental, cross-sectional, and explanatory study, the responses of 518 participants (men and women) who confirmed being consumers of the healthy brands of food were analyzed. The study included adults aged from 18 to 58 years recruited using non-probability sampling. Data was collected using a self-report form and statistically analyzed using Smart PLS. The findings support that health awareness positively and significantly influences perceived quality value, perceived financial value, perceived social value, and perceived emotional value; contrary to this, it was detected that the perceived financial value does not influence the willingness to consume foods from healthy brands. This study contributes significantly to health science by showing how the theory of perceived value predicts the intention to consume healthy brands, with health consciousness intervening in this prediction. Therefore, it is concluded that the study population that consumes healthy foods has experienced the positive impact of perceived value and reports that the factors that comprise it influence their intention to consume healthy foods.

## 1. Introduction

Eating healthy foods is essential to maintain good health and achieve a long quality of life [1,2,3]. The benefits go beyond disease prevention, as they also influence aspects such as energy, mental well-being, and overall quality of life [4,5]. Adopting healthy eating habits is an investment in personal well-being and a key component to a whole and healthy life [6]. Nowadays, people are more aware that a healthy and balanced diet and lifestyle could influence purchasing decisions [7,8]. This is where brands that offer healthy options can appeal to this growing group of health-conscious consumers [1,5,7].

Companies, governments, and international organizations have been joining social efforts to improve people’s lives and health. A clear example of effort is revealed in the formulation of international policies by the United Nations’ (UN) 2030 Sustainable Development Agenda. This is a normative agenda on sustainable development, covering 17 Sustainable Development Goals (SDGs) and 169 goals for achieving them. The OSD—Zero Hunger (SDG 2) and Health and Well-being (SDG 3)—are strongly allied under the mission of achieving a healthy diet for all people. By addressing malnutrition and guaranteeing access to nutritious food, we contribute significantly to improving the health of a population and moving towards sustainable development [9].

There is a strong social trend of great interest in sustainability and conscious eating. Previous studies confirm that this can affect people’s willingness to consume foods from brands that support these trends [9,10]. This topic is a complex phenomenon that involves a combination of individual and contextual factors [11], the same ones that contribute to the formation of a healthier society [5], where consumers can make informed decisions contributing to individual and community well-being. Following this same direction, continuous education on healthy eating is vital to address the public health challenges that each country faces, and consumers with high levels of education will have a healthier consumption philosophy [12].

Health is one of the main drivers of healthy food consumption [13]. Considering this, health consciousness is the tendency to care for one’s health [6]. Although most people’s level of health consciousness is related to eating behavior [14], health-conscious people tend to be willing to do something for themselves and take action for their health. Many studies show a correlation between health consciousness and consuming organic and healthy foods [15]. Most people believe that healthy brand products are good for people’s health because they contain healthier nutritional values than conventional products and are safer. That is, the consumer leans toward foods that help them be healthier without harming the environment [16]. Therefore, people with low health consciousness are often less motivated to engage in behaviors that help them stay healthy [15].

One of the theories applied to these health and nutrition contexts is the theory of perceived value. This theory was initially proposed by Zeithaml and is currently applied in research on perceived value and purchase intention in various contexts [17]. Many studies show that perceived value predicts consumer purchase intention better than satisfaction or quality [17,18]. Perceived value is defined as the general psychological evaluation [17] and subjective nature of a product or service by consumers, measured by their perceptions, generally influenced by benefits and costs [19,20]. Perceived value has been widely used in behavioral research and is closely related to customer loyalty, satisfaction, and continuance intention [21]. In this sense, research shows that perceived value is a key factor influencing consumer attitudes toward purchases [18], which is based, among other things, on past experiences [8].

After carefully reviewing the background, there is evidence of growing interest among academics, business, and health professionals in continuing to research these topics. It can be validated that scientific dissemination has been increasing since 2006, and the interest in continuing research is evident. The preceding studies have been applied to various areas, sectors, and populations, such as management and business, social sciences, medicine, environmental sciences, decision sciences, and psychology. Bibliometric indicators reveal that the ten countries that publish the most scientific results are the USA, China, Taiwan, South Korea, Great Britain, Indonesia, Spain, Malaysia, Australia, and India. When evaluating scientific dissemination by country, it has been found that the research carried out in the Peruvian population needs to be expanded; more scientific literature can support and guide future research in these areas.

Despite the importance that has been given to the present topic, within the scientific literature available, no studies have been found that develop or explain health awareness and its influence on the dimensions of perceived value and, in turn, its influence on the purchase intention of healthy products. Thus, this research aims to fill this knowledge gap by focusing on a Peruvian context and proposing future research that addresses other population scenarios. Furthermore, given the prevalence of malnutrition, diet, and health-related diseases, this study could provide valuable contributions to professionals in academia and related fields.

Considering what was referred to in the previous paragraphs, the objective of the present study is to determine the influence of health consciousness on the dimensions of perceived value and their impact on the willingness to consume foods from a healthy brand. Furthermore, specific objectives are set to identify if (a) health consciousness influences the perceived quality value of a healthy product, (b) health consciousness influences the perceived social value of a healthy product, (c) health consciousness influences the perceived emotional value of a healthy product, (d) health consciousness influences the perceived financial value of a healthy product, (e) perceived quality value influences willingness to consume healthy brand foods, (f) perceived social value influences willingness to consume healthy brand foods, (g) perceived emotional value influences willingness to consume healthy brand foods, and (h) perceived financial value willingness to consume healthy brand foods. Next, this study is divided into the following sections (having already covered the first section): Section 2 contains the theoretical framework and hypothesis development. Section 3 provides materials and methods. Section 4 focuses on the results. Section 5 refers to the discussion, and Section 6 to the conclusions. 

## 2. Theoretical Framework and Hypotheses Development

Health consciousness has increased over time, gaining a notable boost since the COVID-19 pandemic. Given this circumstance, research has emerged that examines customer perception of a brand or product, even more so when it comes to rating key aspects of perceived value, such as quality [8,22]. According to background research, it has been shown that although health consciousness is part of a consumer’s philosophy regarding conscious eating, this philosophy can vary depending on the context of the country and culture. Despite these differences, the consumer tends to evaluate a product and, among other characteristics, its quality [23]. Although the food industry sector is considered unbalanced due to the presence of unsustainable practices, it has been detected that within the framework of satisfying human needs, individuals who intend to maintain a healthy diet maintain practices of health consciousness, thus seeking the quality of the products received [24]. With this, the following hypothesis is proposed:

**H1.** 
*Health consciousness influences the perceived quality of a healthy product.*


Some studies establish that health-conscious consumers, independently of thinking about their well-being, also think about the well-being of others, which is why they evaluate the impact of the brand on society. This attitude is known as social value [25,26]. Likewise, scientific records establish that behaviors that benefit sustainability are part of the perceived social value since multiple positive impacts have been recognized. Here, the consumer is aware of caring for himself and the environment and adopting other behaviors that impact the environment [27]. In this way, the health-conscious consumer limits him/herself to marketing efforts and focuses on choosing a healthy product that cares for his/her environment since he/she recognizes its function as a catalyst for social change [28]. What was previously mentioned leads to the following hypothesis:

**H2.** 
*Health consciousness influences the perceived social value of a healthy product.*


Taking into account that consumer segmentation revolves around specific criteria, it is established that one of these is the preference for food [29,30]. In this way, the literature establishes that some consumer behaviors are closely linked to health consciousness [31]. This is a public health approach that impacts consumer experiences. On the other hand [32,33,34], it describes that consumers who are health conscious are more demanding in their criteria when evaluating certain products since they are also characterized by having a particular interest in knowing it more after choosing it, buying it again and again, and feeling the experience of the benefits the product creates, establishing a deep and sustainable emotional bond and connection over time, which is called emotional value. With this, the following hypothesis is proposed:

**H3.** 
*Health consciousness influences the perceived emotional value of a healthy product.*


In the same way, it has been identified that when consumers are aware of their health, they also look for environmentally friendly products, and when it comes to purchasing them, they are willing to pay as long as it does not exceed the projected budget [35,36]. This means the health-conscious consumer plans ahead and allocates a budget according to consumer preferences regarding healthy products. With this, the following hypothesis is proposed:

**H4.** 
*Health consciousness influences the perceived financial value of a healthy product.*


Perceived quality involves a multidimensional assessment of the attributes of healthy foods from the consumer’s perspective, including beliefs about nutritional value, naturalness, and positive health impacts [15,24,30,37]. Regarding purchase intention, it expresses the subjective probability that consumers will choose to obtain healthy food shortly, determined by confidence in the benefits of the product, personal adherence to sustainability, and perceived social norms [14,16,36,38]. In this way, various studies support that functional and organic foods with high dietary quality increase the probability of future purchases, both in Western and Asian consumers [12,39,40,41]. What was reviewed raises the following hypothesis:

**H5.** 
*Perceived quality influences the purchase intention of healthy products.*


Perceived social value involves the subjective assessment of improvements in status and image within the social circle, which current consumers can obtain by adopting positively validated behaviors in their community [3,22,38,42]. This involves perceptions about optimizing interpersonal relationships, reputation, and self-concept by manifestly adhering to prevailing trends or norms [2,4,43,44]. Some studies indicate that this perceived social value can increase the estimated subjective probability that millennials and centennials will choose food options promoted as healthy by their reference groups, thus increasing adherence to those socially validated consumption behaviors [39,42,43]. With the mentioned above, the following hypothesis is proposed:

**H6.** 
*Perceived social value influences the purchase intention of healthy products.*


Perceived emotional value involves the subjective assessment of feelings of connection and well-being that current consumers in regions of exceptional longevity associate with organic foods [2,21,45,46,47]. Here, the emotional usefulness and symbolic expressiveness of cultural and environmental preservation options are incorporated [48,49,50]. Evidence indicates that the high emotional value perceived in organic products increases the estimated probability of healthy purchasing the following year, increasing conscious eating behaviors [50,51]. Therefore, this hypothesis is proposed:

**H7.** 
*Perceived emotional value influences the purchase intention of healthy products.*


The perceived financial value implies the integrative evaluation of the economic benefits and costs of obtaining a product for the consumer [52,53,54]. In addition, it ranges from quality to price to comfort to perceived savings to satisfying the buyer’s needs [49,55,56]. Therefore, the evidence indicates that this financial value positively impacts the purchase intentions of ecological products in different age groups [38,50,57], promoting conscious consumption behaviors. With this, the following hypothesis is proposed:

**H8.** 
*Perceived financial value influences the purchase intention of healthy products.*


Considering the hypotheses mentioned above, the ensuing conceptual model of the study can be visualized, as depicted in Figure 1.

## 3. Materials and Methods

This article aimed to build an explanatory model through an empirical study to examine healthy consciousness (HC), perceived quality value (PQV), perceived social value (PSV), perceived emotional value (PEV), perceived financial value (PFV), and the willingness to consume healthy brand foods (WCHBF) that takes place in the Peruvian market. The study used a self-administered questionnaire using a quantitative, non-experimental, and cross-sectional design approach [58].

### 3.1. Sample and Procedure

It is important to note that the study population is more representative of young people since they are the ones who maintain a high tendency to consume healthy products [59,60]. For this study, Peruvian residents who stated that they consumed foods from a healthy brand (Union brand) were summoned. The brand in question has been in the Peruvian market for more than 90 years and, since its founding (1929), has sought products with a high nutritional value index that provides, to a certain extent, the prevention of diseases. In addition, being characterized by making healthy products available to the market, allowing a healthy experience for the consumer, this brand is socially responsible by contributing to the educational development of young university students by allowing them to finance their studies through the sale of their products.

One solid condition for participants to take part in this study is that they had to be of legal age and had to have consumed the Union brand. In addition, all participants were informed about the research objectives and the use of the information collected. Before proceeding with filling out the survey, they signed informed consent. Approval was obtained from the school’s ethics committee of Postgraduate Studies of the Universidad Peruana Unión, according to document 2023-CE-EPG-00041. The questionnaire was self-administered and anonymous, thus increasing the probability of obtaining honest answers. The survey was hosted in the Google Forms application and shared through social networks, official academic WhatsApp groups, and official groups of the Union brand. Thus, the participation of 518 individuals selected through non-probabilistic convenience sampling was achieved [61], whose characteristics are detailed in Table 1.

### 3.2. Measurement Scales

The digital questionnaire was structured in 3 sections: The first section presented the instructions for filling out the questionnaire and the informed consent through the statement “I agree to participate”. The second section contained the measurement scales regarding healthy products, specifically the “Union brand”. And the third section was composed of questions related to sociodemographic data such as age, sex, marital status, religion, educational level, and family economic income. The constructs reached a high level of reliability, with Cronbach’s alpha values of 0.951 for health awareness, 0.947 for emotional value, 0.929 for perceived financial value, 0.954 for perceived quality value, and 0.929 for willingness to consume healthy brand foods. Willingness to consume healthy brand foods (WCHBF) was measured in 7 items for each variable [62]. Using the proposal of Köse and Kırcova [13], health consciousness (HC) was assessed using 7 items; however, to measure perceived quality value (PQV), perceived social value (PSV), perceived emotional value (PEV), and perceived financial value (PFV), 3 items were used for each construct (Appendix A). All items were evaluated using a 5-point Likert-type scale, where 1 means “Strongly disagree” and 5 means “Strongly agree”.

### 3.3. Statistical Analysis 

The statistical analysis of this research was performed using a two-step approach: first, the evaluation of the measurement model, and second, the evaluation of the structural model [58]. For this purpose, the statistical software Smart-PLS version 4.0 was used to perform the reliability test of the measurement model, such as discriminant validity and convergent validity [58], and also to test the hypotheses of the structural model. In addition, this study also used IBM SPSS version 25 software to analyze the demographic data of the respondents, which are shown in Table 1.

To determine the influence of health consciousness on the dimensions of perceived value and the influence of these on the willingness to consume foods from a healthy brand (see Figure 1), the partial least squares structural equation model (PLS-SEM) was used [63] since it is a comprehensive approach to multivariate statistical analysis, it involves several variables equal to or greater than three variables, and it includes structural and measurement components to examine the relationships between each of the variables in a conceptual model simultaneously [58].

The measurement instrument evaluation process examined three indicators. (1) Internal consistency and reliability, in which Cronbach’s alpha and the composite reliability (CR) indices were sought to be above 0.70. (2) Convergent validity is met when all items of the construct have loadings greater than 0.70, and the average variance extracted from the construct is greater than 0.50. Moreover, (3) the construct’s discriminant validity is met when the coefficients of the Heterotrait–Monotrait (HTMT) indicator are below 0.85 [64].

To evaluate the structural model, it was first verified whether the relationships established in the model were significant; for this, the *p*-value had to be less than 0.05. Next, two indicators, R2, were examined to test the model’s predictive power. The adjusted R2 values of 0.19, 0.33, and 0.67 are considered weak, moderate, and substantial, respectively [65]. Likewise, in behavioral studies, a value of 0.2 for R2 is acceptable [63].

## 4. Results

The results are presented in two stages: (1) measurement model evaluation, which evaluates the validity and reliability of the measurement model, and (2) structural model evaluation, which evaluates the structural model and addresses the relationships between the constructs [63]. 

### 4.1. Evaluation of the Measurement Model

Regarding the convergent evaluation, Table 2 shows that all the items of the six constructs comply with this validation since all the factor loadings were less than 0.70 [58]. Furthermore, it is confirmed that Cronbach’s Alpha and composite reliability are greater than 0.70. Likewise, it is confirmed that all constructs’ average variance extracted is greater than 0.50. Therefore, the convergent validity of the measurement model was excellent.

To evaluate this study’s discriminant validity, the Heterotrait–Monotrait criterion was used. This criterion is met if the HTMT value is less than 0.95 [64]. Table 3 shows that this criterion is met since the highest value is 0.781, well below the required limit. Therefore, discriminant validity was established between the six constructs of the model.

### 4.2. Structural Model Evaluation

Table 4 and Figure 2 show the results of the structural model with path coefficients between −1 and +1 [58]. The R2 coefficients of the present work for PQV, PSV, PEV, PFV, and WCHBF were 0.333, 0.129, 0.290, 0.285, and 0.615, respectively. That is, the R2 values had values from acceptable to substantial, except for PSV, which is weak. Therefore, the values show that the variables of the present study explain an acceptable percentage of the variance of the WCHB. The overall model fit was measured using the root mean square residual (SRMR), resulting in a value of 0.035 for this indicator, below the recommended threshold value of 0.080 [63].

Hypothesis tests and evaluation of path coefficients can be seen in Table 4. The results show that HC positively and significantly influences PQV, PSV, PEV, and PFV, supporting hypotheses H1, H2, H3, and H4. The results also show that PQV, PSV, and PEV positively and significantly influence WCHBF, which supports hypotheses H5, H6, and H7. This model indicates that PFV does not influence WCHBF, so H8 is not accepted.

## 5. Discussion

The results of this study have shown that health awareness influences the perceived quality of a healthy product. To support this finding, research has been identified that indicates that the greater the health awareness, the higher the evaluation of the product’s quality; this means that the health-conscious consumer usually seeks more information about the quality of the product to be purchased [66,67]. Other studies that support the results shown establish that nutritional knowledge allows consumers to reflect and make the appropriate and informed decision regarding their diet, thus increasing health awareness, a decision that is based on the evaluation of the quality of the food product [67,68] and is that in a scenario where healthy products have attributes associated with higher quality standards, they are recognized and valued with greater emphasis in consumers who maintain health awareness [37,69]. 

According to the results, it has been shown that health awareness influences the perceived social value of a healthy product. This finding is also reported by Sánchez-Feijoo et al. [70], who establish that products from a brand with a social value that also seeks common well-being have a better perspective from the perspective of a health-conscious consumer [44]; these consumers are also the ones who have the greatest susceptibility to opt for those products whose mechanism maintains and promotes social value [71]. In addition, other research that supports the results of this study shows that health-conscious consumers have a high tendency to know the social behavior of the company or brand in the face of various environmental problems; therefore, their behavior tends to validate their commitment to products and/or brands that, independently of meeting quality standards, are aligned with social impact practices [72]; this means that health-conscious consumers, beyond seeking benefits regarding their physical health, also seek a special contribution to society [8].

Another factor that influences health awareness is the perceived emotional value of the healthy product, as demonstrated in this study. In this regard, Zahid et al. [73] demonstrate that affective emotions emerge with greater emphasis when a consumer identifies that the consumption of a specific product generates well-being. That is, when he/she becomes aware of his/her health, then repurchase behavior comes from this, which refers to a scenario where the consumer reduces his/her susceptibility to consider alternative products due to the attachment that has been created [74]. Under this same context and to support the findings of this study, a recent study has been identified that refers to the health-conscious consumer as that character who associates a healthy product with emotional value, and that is that when a consumer believes in a product, and it meets their expectations of health and well-being, an emotional relationship is generated, the same one that derives from a pleasant experience [75,76]. 

Likewise, it has been found that health awareness influences the perceived financial value. This result is completely consistent with the studies of Vieria [77], who established that when a consumer chooses a healthy food, they also evaluate the price [78]. For example, a low price could be a reason for consumer distrust [79]. In this sense, there is research that supports that healthy products must guarantee equity between quality and price since health-conscious consumers seek to obtain nutritional benefits that ensure health care in exchange for a financial value that the product or brand merits [80,81]. In this case, health awareness influences the financial value when evaluating the nutritional benefit against a justified and balanced price.

Another finding establishes that the perception of quality exerts a positive influence on the intention to purchase healthy products. Research that has addressed this topic strongly supports this behavior; according to Ordoñez et al. [27], perceived quality significantly affects purchase intentions. Recent studies [38,39] have shown that quality is a relevant aspect that can affect the purchasing decision when choosing a product or selecting a brand [40]. Under this context, it is necessary that companies develop strategies to communicate quality attributes in order to achieve a true purchase intention for their products. Likewise, there is other research that supports the results of this study by indicating that one of the criteria that drives purchase intention is quality and that this criterion is part of the consumer’s decision when purchasing a product [82,83]. 

Likewise, it has been shown that perceived emotional value influences the purchase intention of healthy products; this finding is supported by researchers Sánchez et al. [84] and Bonisoli and Micolta [85] who place special emphasis on specifying that emotional value is part of a set of evaluative aspects that have a positive influence on the consumer’s purchase intention; furthermore, another recent study has shown that the interaction between emotional value and purchase intention of healthy products is explained by consumer perception; that is, when they feel an important attachment between themselves and the brand and/or product, they will not hesitate to choose it [86]. In this way, emotional value assumes a leading role where the connection between the consumer and the product is lasting, thus generating greater willingness to purchase when choosing a product, which generates an important advantage for the brand over the competition since the barrier of the purchase modality is not a limitation when the emotional value exists [87,88].

Another result, in the applied context, indicates that financial value does not influence the purchase intention of healthy products; this result contradicts some previous research that determines that price influences purchase decisions [89]. However, recent studies have shown that when it comes to healthy products, quality plays a more important role in purchase intention [90,91]. This means that financial value is not necessarily a determinant of purchase intention. Furthermore, other research reveals that nowadays, consumers prioritize their health and are willing to pay a high price if required [92,93]. With this recent background, the results found in this study are supported, taking into account that the financial value can be a decisive element depending on the context where it is applied, being in the context of healthy foods a secondary factor that does not determine the purchase intention. 

In general terms, evidence has been found that health awareness influences three dimensions of perceived value (perceived quality, social value, emotional value). This means that every health-conscious consumer has a high tendency to value the quality of health products more, granting greater social value to products due to the recognition they have in the social environment, which means they experience a positive emotional value. These multiple influences highlight how important it is for consumers to become health conscious, thus avoiding negative outcomes regarding their health [94,95]. With this, companies could establish clear policies that lead to capturing and achieving loyalty in a segmented manner with customers that are health-conscious consumers while having support from governments and health authorities, who maintain an important participation in the improvement of public health conditions [96,97].

### 5.1. Theoretical and Practical Implications

Despite recognition that a healthy diet is economically fair, affordable, nutritionally adequate, and safe, few countries have adopted healthy eating principles in government dietary recommendations. Based on this study’s findings, marketing strategies can be developed to increase consumer preference for healthy branded products. To this effect, it is necessary to promote consumer education so that they can have the ability to assertively decide which healthy products to consume. Under this circumstance, companies are called to use attractive packaging that draws the attention of the consumer by seeking the collaboration of influencers to promote healthy eating.

Given the critical role that consumers’ health consciousness plays in healthy food choices, marketers and government policymakers should focus on conducting educational campaigns through various media channels to promote awareness about health topics among consumers. In educational programs presented on various media and social media channels, recognition of opinions from important leaders can effectively change consumer attitudes and dietary choices. In addition to providing knowledge about healthy foods, marketers can emphasize the psychological benefits of nutritious foods during the consumption experience. Since food consumption is experiential and primarily related to consumers’ emotions, the psychological benefits associated with the consumption experience will increase consumers’ willingness to purchase foods from healthy brands.

Alternatively, once this research is conducted, companies could source food products tailored to the preferences and values of health-conscious consumers. If they offer health benefits, they could also innovate by creating new ingredients and recipes low in sodium, saturated fat, and added sugars.

Through market segmentation, food industries can easily find a market segment that consists of health-conscious consumers willing to consume products from a healthy food brand. Therefore, they can easily change production and marketing according to consumer interests and preferences. They may even have a different product for health-focused consumers. Likewise, they can collaborate with the health industry, nutritionists, or wellness experts to be the recommended brand. Thus, brand credibility will increase. It is also believed that restaurant companies can consider these options and offer healthier alternatives to attract consumers from this segment. In this way, they can improve their brand’s reputation by demonstrating a complete and genuine commitment to the health and well-being of consumers.

Finally, this study can provide valuable information for companies to adapt their strategies, policies, products, and operations to meet market demands aimed at population health. Adapting to the desires of health-conscious consumers improves brand image, reputation, and perceived value and contributes to the general well-being of society.

### 5.2. Limitations and Future Research 

Although this study provides an in-depth understanding of how perceived value theory predicts willingness to consume foods from a healthy brand and how health consciousness may play a role in this prediction, there are limitations to the research. For example, although we have information on the economic income of the study participants, an analysis of the association of this income with the purchase of healthy products has not yet been carried out. The reason for this is that homogeneous samples were not obtained that would allow comparison. Under this context, it is proposed that future research investigate whether the economic condition of individuals could be associated with the willingness to consume foods from a healthy brand.

Another limitation of this study corresponds to the sample that was selected. It is a non-probabilistic sampling that was used at the convenience of the researcher. As a result of this, a high representation of women was obtained in the surveys, with a high number of participants whose ages range between 18 and 25 years and a significant percentage of participants who declared themselves to be Adventists (a group characterized by having a differentiated lifestyle with respect to their diet, customs, and beliefs), which means this could be a bias regarding the results obtained. The lack of balance between the number of men and women could reduce the possibility of generalizing the results. Considering this, it is proposed that future studies segment the sample in order to include a greater diversity of participants and carry out comparative studies between sociodemographic characteristics such as religious belief, age, gender, and marital status.

On the other hand, the concept and knowledge of each study participant about healthy brands have not been recorded; this could represent research bias because the lack of this detail could vary respondents’ perceptions. Thus, it is proposed that future investigations carry out a quasi-experimental investigation where, in the first instance, the study population is made aware of the consumption of healthy foods, thus ensuring that the participants have prior information regarding healthy foods.

Furthermore, other research could focus on qualitative studies that explore motivations or barriers that influence an individual’s consumption decision when choosing healthy foods. The results found would be key information to promote the choice of healthy foods and increase awareness of health.

## 6. Conclusions

This study shows how the theory of perceived value can predict the willingness to consume foods from a healthy brand, with health consciousness intervening in this prediction. Thus, it is concluded that the population under study that consumes healthy foods has experienced the positive impact of perceived value and reports that the factors that comprise it influence their intention to consume nutritious foods. In this case, it is important that healthy brand companies can consolidate and strengthen quality and social, emotional, and financial value. This fact constitutes the option of activating two agents of change that can contribute to the willingness to consume food. First, educational institutions could be part of the change in consumer behavior since educational programs can promote health consciousness, encouraging students to choose healthy foods in their daily diet. On the other hand, there are also companies dedicated to producing healthy foods; these assume a fundamental role by having the capacity to directly impact the intention to consume healthy foods by consolidating the value of perceived components. Focusing on the components of perceived value is part of a social initiative that aims to influence the Peruvian population’s habits positively. 

Likewise, the need to disseminate healthy food consumption patterns is conclusive. For this purpose, actions must be taken to transform consumers’ attitudes and behaviors so that they are willing to integrate alternatives that lead to healthy eating. This fact is noted in a health-conscious approach. 

Finally, considering that this research’s findings indicate that health consciousness can influence perceived value, the need to establish strategies that place high relevance on health consciousness is highlighted, extending from individual to collective decisions. For this reason, it is necessary to integrate educational programs and health lectures that allow the population to make better decisions regarding their diet.

## Figures and Tables

**Figure 1 nutrients-16-01995-f001:**
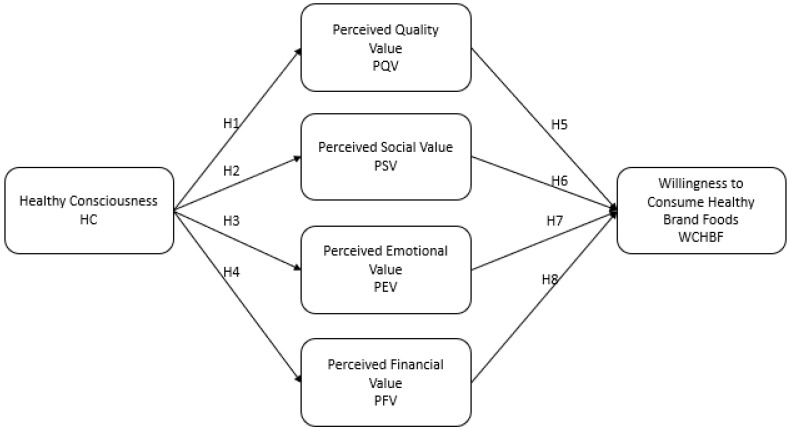
Proposed model.

**Figure 2 nutrients-16-01995-f002:**
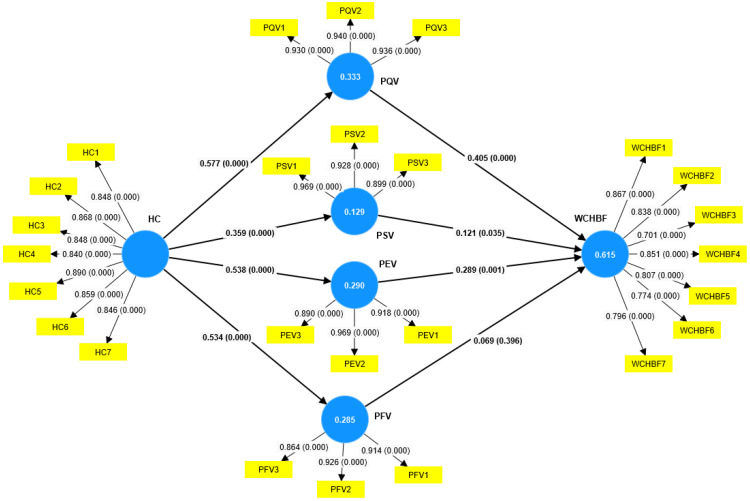
Structural model.

**Table 1 nutrients-16-01995-t001:** Sociodemographic characteristics (*n* = 518).

Category	Frequency	Percentage
Age		
18–25	451	87.0
26–33	41	7.9
34–42	14	2.7
43–50	6	1.2
51–58	6	1.2
Gender		
Male	183	35.3
Female	335	64.7
Marital status		
Married	30	5.8
Divorced	2	0.4
Single	486	93.8
Religion		
Adventist	444	85.7
Catholic	57	11
Evangelical	7	1.4
Other	10	1.9
Academic formation		
Secondary completed	19	3.7
Advanced technician	4	0.8
University (undergraduate)	462	89.2
University (postgraduate)	33	6.3
Family economic income		
Up to 2 minimum salaries	266	51.4
From 3 to 4 minimum salaries	131	25.3
From 5 to 10 minimum salaries	92	17.8
From 11 to 20 minimum salaries	19	3.6
Greater than 20 minimum salaries	10	1.9

**Table 2 nutrients-16-01995-t002:** Scale elements.

Predictor	Code	Outer Loadings	α	Composite Reliability (rho_a)	Composite Reliability (rho_c)	AVE
Healthy consciousness (HC)	HC1	0.833	0.951	0.951	0.960	0.773
	HC2	0.893				
	HC3	0.896				
	HC4	0.907				
	HC5	0.897				
	HC6	0.870				
	HC7	0.781				
Perceived emotional value (PEV)	PEV1	0.943	0.947	0.949	0.966	0.904
	PEV2	0.953				
	PEV3	0.925				
Perceived financial value (PFV)	PFV1	0.891	0.929	0.930	0.955	0.875
	PFV2	0.929				
	PFV3	0.907				
Perceived quality value (PQV)	PQV1	0.945	0.954	0.954	0.971	0.917
	PQV2	0.920				
	PQV3	0.943				
Perceived social value (PSV)	PSV1	0.963	0.952	0.953	0.969	0.912
	PSV2	0.953				
	PSV3	0.941				
Willingness to consume healthy brand food (WCHBF)	WCHB1	0.800	0.929	0.931	0.943	0.701
	WCHB2	0.778				
	WCHB3	0.779				
	WCHB4	0.778				
	WCHB5	0.826				
	WCHB6	0.809				
	WCHB7	0.785				

**Table 3 nutrients-16-01995-t003:** Herotrait–Monotrait (HTMT) matrix.

	HC	PEV	PFV	PQV	PSV	WCHB
HC						
PEV	0.538					
PFV	0.534	0.781				
PQV	0.578	0.742	0.754			
PSV	0.359	0.646	0.612	0.455		
WCHB	0.561	0.721	0.673	0.724	0.535	

**Table 4 nutrients-16-01995-t004:** Estimates of the proposed hypotheses.

H	Hypothesis	Original Sample (O)	Sample Mean (M)	Standard Deviation (STDEV)	T Statistics (|O/STDEV|)	*p* Values	Decision
H1	HC -> PQV	0.577	0.577	0.039	14.829	0.000	Accepted
H2	HC -> PSV	0.359	0.360	0.049	7.312	0.000	Accepted
H3	HC -> PEV	0.538	0.539	0.044	12.284	0.000	Accepted
H4	HC -> PFV	0.534	0.534	0.042	12.673	0.000	Accepted
H5	PQV -> WCHB	0.405	0.407	0.071	5.693	0.000	Accepted
H6	PSV -> WCHB	0.121	0.122	0.057	2.114	0.035	Accepted
H7	PEV -> WCHB	0.289	0.288	0.086	3.375	0.001	Accepted
H8	PFV -> WCHB	0.069	0.068	0.082	0.848	0.396	Rejected

## Data Availability

Data are available on request from the authors.

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
