# Peer review of "Using the Theory of Perceived Value to Determine the Willingness to Consume Foods from a Healthy Brand: The Role of Health Consciousness"

_nutrients, 2024, doi:10.3390/nu16131995_

Round 1

Reviewer 1 Report

Comments and Suggestions for Authors

The manuscript addresses an important topic and the presentation for the most part is strong.

A major concern, however, is the diffculy generalizing the findings owing to a non-representative dataset. Without randomization it is important to address issues such as the overrepresentation of female respondents through post-stratificaton or other such methods. The authors should try to replicate the findings with re-balanced data and comment on any discrepancies.

Implications of the findings for practice need to be explicated further, and the authors should add a paragraph or so of discussion about implications for policy.

Comments on the Quality of English Language

Although the mansucript is readable, occasional passages will require careful editing.

Author Response

Dear Reviewer

We extend our sincere gratitude for your insightful comments, which have been invaluable in enhancing the quality of our manuscript. Your thoughtful feedback has contributed significantly to refining our work, and we have made concerted efforts to address each of your suggestions.

We are optimistic that this revised version of the paper now meets the anticipated standards for publication in this esteemed journal. Below is a comprehensive list of responses addressing your comments and suggestions.

Thank you once again for your time and expertise.

Best regards,

Reviewer 2 Report

Comments and Suggestions for Authors

Dear authors,

It would be advisable for the authors to include a sentence about the main results in the abstract. After the methodology, the abstract continues directly with the study contributions and conclusions. 

The introduction starts off very well, with the authors setting out the general context of the problem being studied. Unfortunately, the authors fail to outline what the identified gap in the scientific literature is and how they propose to fill it. Also, the aim of the research is not defined, only the objective. It is very general. The authors should formulate an aim from which they derive specific objectives. At the end of the introduction, it is advisable for the authors to present the structure of the paper.

The hypotheses are correctly formulated and are well substantiated with studies from the scientific literature, and the proposed model is well described.

How can the authors be sure that the people who completed the questionnaire have consumed the healthy brand? Are they relying solely on their honest statement?

From a methodological point of view, the authors should describe in detail the research instrument, i.e. the questionnaire.

The authors should explain that they used the snow ball sampling method and explain the limitations of using this method. Authors are also requested to mention in the paper how the link to the questionnaire administered online was distributed and disseminated.

The sample is also extremely unbalanced in terms of the age of respondents, with 87% between 18 and 25 years old. It is also known that the Adventist Church has a set of strict rules. Adventists forbid alcohol, tobacco, coffee, unhealthy food, dancing and marrying people of other religions. Looking at the structure of the sample, 85.7% of respondents are Adventists. How could the results be extrapolated to the entire population?

The statistical analyses carried out should be presented in much more detail.

The results of the analyses are nicely presented, but insufficiently discussed and compared with the results of peer-reviewed studies.

The study has many more limitations than those presented by the authors.

The study has major shortcomings and should be rethought and rewritten.

Author Response

(The authors gave the same response as above.)

Round 2

Reviewer 1 Report

Comments and Suggestions for Authors

The manuscript is much improved.

Comments on the Quality of English Language

Engish is fine. As with all docments, a last-look edit could catch some potential changes.